# Magnetically mediated hole pairing in fermionic ladders of ultracold atoms

Sarah Hirthe[1,2 ✉], Thomas Chalopin[1,2], Dominik Bourgund[1,2], Petar Bojović[1,2], Annabelle Bohrdt[3,4], Eugene Demler[5], Fabian Grusdt[2,6,7], Immanuel Bloch[1,2,7] & Timon A. Hilker[1,2]

Conventional superconductivity emerges from pairing of charge carriers—electrons or holes—mediated by phonons[1]. In many unconventional superconductors, the pairing mechanism is conjectured to be mediated by magnetic correlations[2], as captured by models of mobile charges in doped antiferromagnets[3]. However, a precise understanding of the underlying mechanism in real materials is still lacking and has been driving experimental and theoretical research for the past 40 years. Early theoretical studies predicted magnetic-mediated pairing of dopants in ladder systems[4–8], in which idealized theoretical toy models explained how pairing can emerge despite repulsive interactions[9]. Here we experimentally observe this long-standing theoretical prediction, reporting hole pairing due to magnetic correlations in a quantum gas of ultracold atoms. By engineering doped antiferromagnetic ladders with mixed-dimensional couplings[10], we suppress Pauli blocking of holes at short length scales. This results in a marked increase in binding energy and decrease in pair size, enabling us to observe pairs of holes predominantly occupying the same rung of the ladder. We find a hole–hole binding energy of the order of the superexchange energy and, upon increased doping, we observe spatial structures in the pair distribution, indicating repulsion between bound hole pairs. By engineering a configuration in which binding is strongly enhanced, we delineate a strategy to increase the critical temperature for superconductivity.

Unconventional superconductivity in materials such as heavy fermion systems[11], iron pnictides[12], layered organic materials[13], cuprate superconductors[14] and twisted bilayer graphene[15], arises in the vicinity of magnetically ordered states. A common mechanism consisting of dopant pairing mediated by magnetic fluctuations is thus believed to be at the heart of these superconducting states[2], but a detailed understanding of the underlying physics remains a key problem in quantum many-body physics.

A promising tool to explore strongly correlated quantum systems is analogue quantum simulation using ultracold atoms[16]. Recent experimental progress investigating doped antiferromagnets has been made using single-site resolved fermionic quantum gas microscopes[17–22], which provide microscopic real-space correlations complementary to the spectroscopic and transport measurements performed in solids. They typically simulate the Fermi−Hubbard model, consisting of itinerant spin-1/2 fermions within a single band of a periodic lattice. Even though the two-dimensional Fermi−Hubbard model displays many characteristics also found in the high-temperature (high-$T_c$) superconducting cuprates[3], the existence of pairing and superconductivity in this model remains a subject of debate[23,24].

To shed light on the pairing mechanism in doped Mott insulators, several theoretical studies considered doped Fermi−Hubbard and $t$−$J$ ladders (see Eq. (1))[4–6,8,25], in which accurate numerical solutions can be obtained using the density-matrix renormalization group (DMRG) method[26]. In solid-state experiments, ladder materials have also been shown to display superconductivity[27–29]. A paradigmatic case for theoretical investigation, which exhibits large binding, is the regime in which interchain magnetic exchange is larger than single-particle interchain hopping[7]. These parameters could, however, not be justified microscopically for condensed matter systems and are unphysical in the framework of a pure Fermi−Hubbard system (Methods). The key motivation for our work was to provide an experimental realization of this system that has been considered only a theoretical abstraction. We achieve this by extending Fermi−Hubbard ladders at large interactions with a potential offset, effectively realizing a mixed-dimensional (mixD) system[10].

The essential physics of our experiment is captured (Methods) by the $t$−$J$ Hamiltonian

$$\hat{\mathcal{H}} = \sum_{\langle i,j \rangle, \sigma} \hat{\mathcal{P}}(-t_{ij}\,\hat{c}^{\dagger}_{i,\sigma}\hat{c}_{j,\sigma} + \text{h.c.})\hat{\mathcal{P}} + \sum_{\langle i,j \rangle} J_{ij}\left(\hat{\mathbf{S}}_i \cdot \hat{\mathbf{S}}_j - \frac{\hat{n}_i \hat{n}_j}{4}\right), \qquad (1)$$

where $\hat{\mathcal{P}}$ projects to the subspace without double occupancies; the hopping energy is $t_{ij} = t_{\parallel}$ ($t_{\perp}$) and the superexchange energy is $J_{ij} = J_{\parallel}$ ($J_{\perp}$)

[1]Max-Planck-Institut für Quantenoptik, Garching, Germany. [2]Munich Center for Quantum Science and Technology, Munich, Germany. [3]Department of Physics, Harvard University, Cambridge, MA, USA. [4]ITAMP, Harvard-Smithsonian Center for Astrophysics, Cambridge, MA, USA. [5]Institute for Theoretical Physics, ETH Zurich, Zurich, Switzerland. [6]Department of Physics and Arnold Sommerfeld Center for Theoretical Physics (ASC), Ludwig-Maximilians-Universität, Munich, Germany. [7]Department of Physics, Ludwig-Maximilians-Universität, Munich, Germany. ✉e-mail: sarah.hirthe@mpq.mpg.de

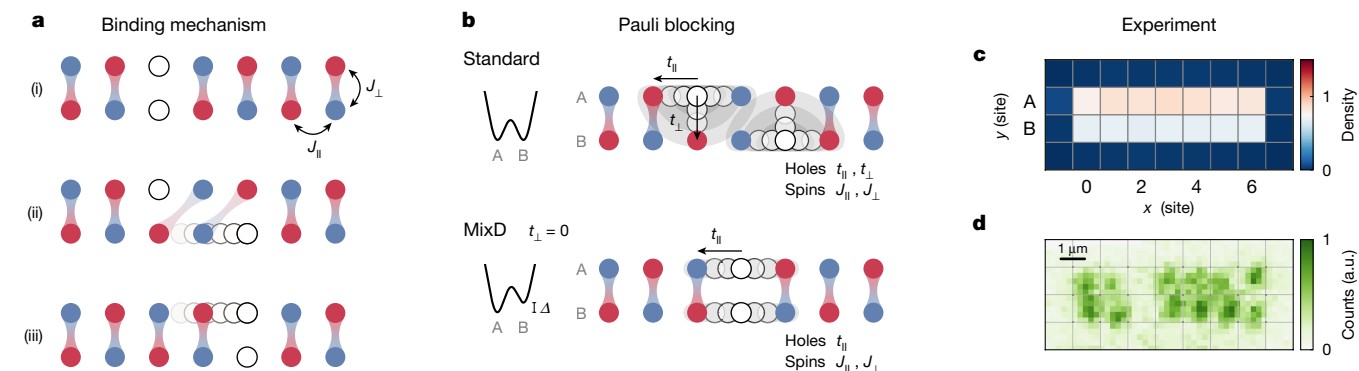

**Fig. 1 | Hole pairing in mixD ladders. a**, Binding mechanism in the $t$–$J$ ladders. Depicted are ladder systems with spin exchange $J_\perp \gg J_\parallel$ that form strong singlet bonds along the rungs. When a single hole from (i) moves through the system, as illustrated in (ii), it breaks the spin order by displacing the singlet bonds. (iii) The magnetic energy cost can be avoided if the second hole restores the spin order by moving together with the first hole. **b**, Pauli blocking of holes. Owing to their fermionic nature, holes repel each other along all directions according to the tunnelling amplitudes $t_\perp$ and $t_\parallel$. Close-distance hole pairs are thus energetically unfavourable. In mixD systems, a potential offset $\Delta$ between the two legs suppresses tunnelling $t_\perp$ and Pauli repulsion only occurs along the legs. Holes on the same rung can thus benefit from the binding mechanism, forming tightly bound pairs with a large binding energy. **c**, Average density of the mixD $L = 7$ ladder system with $\Delta \approx U/2$. **d**, Single experimental shot with two holes on the same rung, exemplifying the bunching of holes in the mixD system. a.u., arbitrary units.

for nearest-neighbour sites $i, j$ on the same leg (rung). $\hat{c}^\dagger_{i,\sigma}$ ($\hat{c}_{i,\sigma}$) denotes the creation (annihilation) operator of a fermion on site $i$ with spin $\sigma = \uparrow, \downarrow$; $\hat{\mathbf{S}}_i$ and $\hat{n}_i$ are the on-site spin and density operators and h.c. means Hermitian conjugate. The mixD system is described by equation (1) for $t_\perp = 0$, which we realize by suppressing tunnelling, but not spin exchange, using a potential offset along the rungs. We also realize the bare Fermi–Hubbard system without additional potential offset and $t_\perp > J_\perp$ (in the following this is denoted as a standard ladder).

The pairing of holes in a model exhibiting only repulsive interactions[30,31] can be understood from the competition of hole delocalization and spin order along the rungs, as illustrated in Fig. 1a. If a hole moves through the system, it displaces these spins creating an energetically unfavourable magnetic configuration. Therefore, a single hole becomes dressed by a cloud of disturbed correlations and forms a polaron with reduced mobility[19,20]. If a second hole moves along with the first hole, it can restore the order in the spin sector, causing the two holes to form a highly mobile bound pair.

However, this process only dominates in the mixD case, whereas in the standard ladders $t_\perp$ is the dominating energy scale. In the latter, binding competes with the repulsion of holes due to Pauli blocking, rendering tight pairs energetically unfavourable (Fig. 1b). Pairing between holes can still occur, but only with a small binding energy $E_b \ll J_\perp$[32,33] and at large pair sizes[34]. By realizing mixD ladders, we strongly suppress the Pauli repulsion between holes along the rung (Fig. 1b), thus engineering a system with strong hole attraction. This binding mechanism is protected by the spin gap and thus persists up to high temperatures of the order $J_\perp$.

We realize ladders of length $L = 7$ in our Fermi-gas microscope with independently tunable optical lattices and single-site resolved optical potential shaping. The mixD $t$–$J$ system is derived as an effective model from fermionic atoms in a two-leg ladder-shaped lattice potential described by the Hubbard parameters $U$ (on-site interaction), $\tilde{t}_\parallel$, $\tilde{t}_\perp$ (tunnelling amplitudes) and with a potential offset $\Delta$ between the two legs. For large $U/\tilde{t}_\parallel$, $U/\tilde{t}_\perp$, the system is effectively described by the $t$–$J$ model and its distinct parameters as in equation (1). For $U > \Delta > \tilde{t}_\perp$, tunnelling along the rungs is suppressed to $t_\perp = 0$, whereas tunnelling along the leg is unaffected, $t_\parallel = \tilde{t}_\parallel$, and spin exchange is given by $J_\perp = 2\tilde{t}_\perp^2/(U + \Delta) + 2\tilde{t}_\perp^2/(U - \Delta)$ (refs. [35,36]) and $J_\parallel = 4\tilde{t}_\parallel^2/U$.

Our experiment begins by preparing a balanced mixture of the lowest two hyperfine states of $^6$Li, which we load into an engineered ladder geometry similar to our previous work[37]. We first load atoms into uncoupled legs of equal potential and then apply an optical

potential offset $\Delta$ to one of the legs using light shaped by a digital micromirror device (DMD). We subsequently connect both legs by slowly lowering the lattice potential in the rung direction (for details see Methods). The offset between the two legs prevents the atoms from tunnelling along the rungs, such that we end up with roughly equally populated legs. Occupation readout is achieved with single-site spin and charge resolution[38]. To highlight the influence of Pauli repulsion on hole pairing, we compare the mixD $\Delta \approx U/2$ case with the standard ladder system at $\Delta = 0$. Both systems are realized in the strong rung-singlet regime, with $J_\perp/J_\parallel = 21(5)$ in the mixD case with enhanced spin exchange and $J_\perp/J_\parallel = 16(3)$ in the standard ladders, where numbers in parentheses denote the uncertainty on the last digit. The mixD system is governed by the energy scales of rung spin exchange and leg tunnelling with $t_\parallel/J_\perp = 0.7(1)$. The strong rung coupling keeps the spatial extent of hole pairs small while staying in the regime in which spin correlations and hole motion compete on a comparable energy scale. For larger $t_\parallel$, the binding energy is even expected to grow[10] but becomes harder to observe in systems of limited sizes because of increased pair sizes.

Figure 1c shows the average density of the mixD system. Quantum fluctuations, which are biased towards the lower leg, as well as preparation imperfections lead to a small residual density imbalance between the two legs. For the data analysis (except Fig. 1c) we only take into account ladders without double occupancies (doublons), such that holes arising from doublon–hole fluctuations do not contribute to our observations. Unless otherwise mentioned, we furthermore restrict the data to realizations with two to four holes per ladder and limit the occupation imbalance between the legs to one hole.

To probe the pairing of holes in our system, we evaluate the hole–hole correlator

$$g_h^{(2)}(\boldsymbol{r}) = \frac{1}{\mathcal{N}_{\boldsymbol{r}}} \sum_{i-j=\boldsymbol{r}} \left( \frac{\langle \hat{n}_i^h \hat{n}_j^h \rangle}{\langle \hat{n}_i^h \rangle \langle \hat{n}_j^h \rangle} - 1 \right),$$

with normalization $\mathcal{N}_{\boldsymbol{r}}$ the number of sites $\boldsymbol{i}, \boldsymbol{j}$ at distance $\boldsymbol{r}$ and $\hat{n}_i^h$ the hole-density operator at position $\boldsymbol{i}$. The function $g_h^{(2)}(\boldsymbol{r})$ is a connected two-point density correlator that is negative if the presence of a hole at position $\mathbf{i}$ makes it less likely to find a second hole at distance $\boldsymbol{r}$ and positive if it makes it more likely. The correlator is bounded by $-1 \le g_h^{(2)}(\boldsymbol{r}) \le (1/n_h - 1)$ with hole density $n_h = N_h/2L$, where $N_h$ is the number of holes in the system.

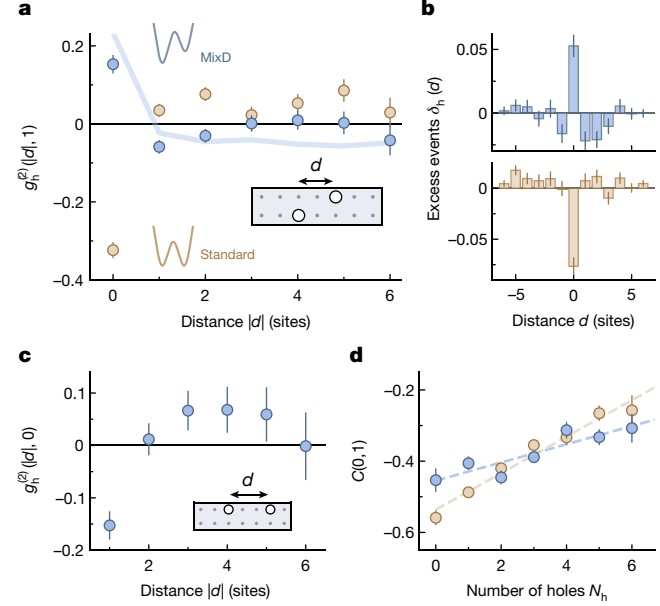

**Fig. 2 | Hole pairing in mixD versus standard ladders. a**, Hole–hole correlator $g_h^{(2)}(d, 1)$ between sites on opposite legs (as illustrated in the inset) for mixD (blue) and standard (brown) ladders with two to four holes per ladder. The strong correlation at $d = 0$ corresponds to two holes on the same rung. Correlations at this distance are strongly enhanced in the mixD system (pairing) and strongly suppressed in the standard ladders (repulsion). The blue line is calculated using MPS at finite temperature $k_B T = 0.8 J_\perp$ and corrected by the experimental detection fidelity (Methods). **b**, Excess events $\delta_h(d)$ of the same data, that is, the likelihood of finding holes at distance $d$ compared with the infinite-temperature distribution. **c**, Hole–hole correlation on the same leg $g_h^{(2)}(d, 0)$ in the mixD system, showing that holes repel each other within the same leg. A finite-size offset correction has been applied to this subfigure (Methods). **d**, Spin–spin correlations $C(0, 1)$ for spins on the same rung depending on the number of holes in the system. The lines represent linear fits. The larger slope indicates that the spin order of the standard system (brown) is more strongly disturbed by holes than the spin order of the mixD system (blue), where paired holes leave the spin order largely unperturbed. The error bars denote one s.e.m. and are smaller than the marker when not visible. Error bars in **a**, **c** and **d** are estimated using bootstrapping.

By evaluating the correlation for holes in opposite legs $\boldsymbol{r} = (d, 1)$, we observe a strong positive signal at distance $|d| = 0$ in the mixD system of $g_h^{(2)}(0, 1) = 0.15(2)$. This corresponds to two holes bunching on the same rung (Fig. 2a). The fast decrease of correlations for $|d| > 0$ indicates that the holes are in a tightly bound state. We find a minimum at $|d| = 1$, which we attribute to the effect of additional holes in the system. These holes are repelled from the hole pair (see also Fig. 4), leading to the weak modulation at larger distances which dominates over the extent of the hole pair at short distances.

By contrast, in the standard ladders of $t_\perp > J_\perp$, a strong repulsion of holes from the same rung is the dominant feature leading to a negative $g_h^{(2)}(0, 1)$ (Fig. 2a). This shows that tightly bound pairs are energetically unfavourable in the standard ladder system.

The attraction (mixD) and repulsion (standard) of holes are also visible in the occurrences of hole distances. In Fig. 2b we plot the histograms of holes found at a mutual distance $d$ as described by $\delta_h(d) = \sum_{i-j=(d,1)} (\langle \hat{n}_i^h \hat{n}_j^h \rangle - n_h^2)$, where subtracting the global hole density $n_h$ removes the uncorrelated distribution. These excess events $\delta_h(d)$ can be interpreted as the likelihood of the hole distance $d$ occurring beyond the probability of a random distribution.

The hole–hole correlator on the same leg (Fig. 2c) shows the effect of hole mobility in the mixD system. It exhibits a minimum at nearest neighbours caused by the Fermi repulsion of holes due to the leg tunnelling being proportional to $t_\parallel$ and a broad maximum around $|d| = 4$. This is the largest mutual distance the two holes can assume without occupying the edge of the system, which is energetically expensive owing to the hard walls blocking hole movement.

We investigate the magnetic origin of the pairing mechanism using the spin correlator

$$C(\boldsymbol{r}) = \frac{1}{\mathcal{N}_r} \sum_{\boldsymbol{i}-\boldsymbol{j}=\boldsymbol{r}} 4(\langle \hat{S}_i^z \hat{S}_j^z \rangle_s - \langle \hat{S}_i^z \rangle_s \langle \hat{S}_j^z \rangle_s) \tag{2}$$

where $\langle \rangle_s$ denotes the expectation value for singly occupied sites. In the doped mixD system we find strong nearest-neighbour spin correlations along the rung of $C(0, 1) = -0.38(1)$, indicating a high singlet fraction, as well as a significant coupling of these bonds along the leg with nearest-neighbour spin correlations of $C(1, 0) = -0.10(1)$. Resolving the rung spin correlations through the number of holes in the system shows a decrease in correlation strength with growing hole number (Fig. 2d). This can be explained by unpaired holes breaking singlet bonds because of their mobility along the ladder. The standard system, which does not display pairing, shows a more rapid loss of correlation strength compared to the mixD system, in which a significant fraction of holes is bound in pairs. This behaviour is directly related to the magnetic origin of pairing, as illustrated in Fig. 1. We attribute the lower correlation strength of the mixD system in the absence of holes to heating caused by the application of the tilt. The much stronger effect of hole doping on spin correlations is, however, unaffected by a small difference in temperature. The slight alternating behaviour of $C(0, 1)$ at low hole numbers in the mixD system, in particular the strong spin correlations for two holes, is reminiscent of the low-temperature behaviour for which rung pairs are dominant over thermal excitations and only odd hole numbers lead to unpaired holes (Supplementary Information).

We estimate the binding energy of the paired state in the mixD system from the experimental data, by comparing it with an analytically tractable effective Hamiltonian. The approach is based on the assumption that the system is reasonably close to the uncoupled rung limit and the bound state can thus be described by two holes on the same rung (for details on the calculation, see Methods). Using the measured and detection-fidelity-corrected probability to find a rung pair and our estimated temperature of $k_B T = 0.77(2) J_\perp$ (Methods), we find an experimental binding energy of

$$E_b = 0.82(6) J_\perp.$$

This is consistent with DMRG calculations giving a binding energy of $E_b^{theo} = 0.81 J_\perp$ and boosts the binding energy by an order of magnitude compared with standard ladders at the same interaction strength and with symmetric coupling (see also Methods). Such a substantial increase in pairing strength is an essential ingredient in the quest for higher temperature superconductivity.

To gain a better understanding of the system dependencies, we probe the influence of temperature and doping on the rung pairing strength. In our experimental regime, nearest-neighbour spin correlations along the rung stand in strong direct relation with the temperature of the system and can therefore be used as an effective thermometer. A mapping between the two is obtained by comparing average nearest-neighbour spin correlations to finite-temperature matrix product state (MPS) calculations (Methods). We observe that the rung hole–hole correlation $g_h^{(2)}(0, 1)$ increases with increasing spin correlation strength (Fig. 3a), and the onset of pairing occurs in the experimental system around temperatures of the order of the spin exchange $J_\perp$. The highest correlations reached, $g_h^{(2)} = 0.3(1)$, are still smaller than the theoretically achievable values of $g_h^{(2)} > 1.2$ for very low temperatures, showing the temperature limitations of the experiment. The repulsion of the

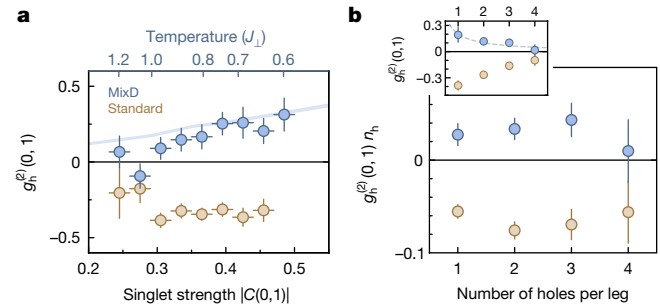

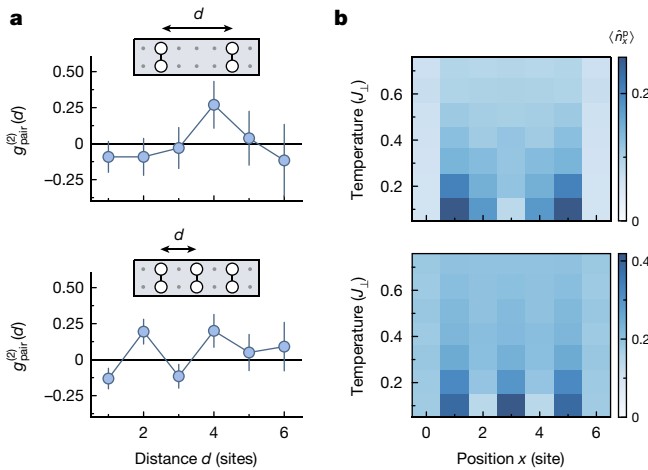

**Fig. 3 | Temperature and doping dependence of hole pairing. a,** Rung hole–hole correlation $g_h^{(2)}(0,1)$ for the mixD (blue) and standard (brown) ladders binned by the rung spin correlations $C(0,1)$ of the system. The temperature of the mixD system (top axis) is estimated by comparing the spin correlations (lower axis) with the theoretical values. The solid line is calculated using MPS and is corrected by the experimental detection fidelity. We see unbinding of pairs at low singlet strength, that is, high temperature. **b,** The hole correlator scaled with the hole density $g_h^{(2)}(0,1)\,n_h$ depending on the number of holes per leg for the mixD (blue) and standard (brown) ladders. Within our error bars, we find the hole binding to be independent of doping. The inset shows the correlator $g_h^{(2)}(0,1)$, where the dashed line is a fit with the inherent $1/n_h$ scaling of the correlator. Error bars denote the bin width of the spin correlations (**a**) and the s.e.m. of the correlator (**a** and **b**).

**Fig. 4 | Distribution of rung hole pairs in the mixD system. a,** Measured pair–pair correlation $g_{pair}^{(2)}(d)$ of rung hole pairs in the experimental system. The upper plot shows the pair–pair correlation for four to five holes, that is, up to two pairs, in the system. The lower plot shows the pair–pair correlation for six to seven holes, that is, up to three pairs in the system. A finite-size offset correction has been applied to the curves (Methods). Error bars were estimated using bootstrapping. **b,** Theoretical (MPS) results for the density of rung pairs in the system for temperatures from $0.1J_\perp$ to $0.7J_\perp$. The upper plot shows the pair density for four holes. The lower plot shows the pair density for six holes. In both cases, the pairs maximize their respective distance, while also avoiding the edge of the system.

standard system remains mostly constant in our temperature regime, as it is governed by the energy scale $t_\perp \gg J_\perp, k_B T$.

To explain the effect of doping on the pair binding, we adjust the correlator $g_h^{(2)}$ to compensate for the intrinsic density-dependent $1/n_h$ scaling. The resulting correlator $g_h^{(2)} n_h$ does not significantly change on increasing the number of holes per leg (Fig. 3b). This is in agreement with a system of independent pairs for the mixD system. For the standard system, hole repulsion is determined by the tunnelling strength $t_\perp$ and is therefore also not significantly influenced by density.

We further investigate the behaviour of several pairs, as their interplay is a key aspect for the competition between superconductivity and charge (density) order[23,39]. For simplicity, we identify the bound state as two holes occupying the same rung. We thus define the pair operator $\hat{n}_x^p$ which is equal to 1 if both sites of rung $x$ are occupied by a hole, and 0 otherwise. Pair interactions are then quantified by the pair–pair correlator

$$g_{pair}^{(2)}(d) = \frac{1}{\mathcal{N}_d} \sum_x \left( \frac{\langle \hat{n}_x^p \hat{n}_{x+d}^p \rangle}{\langle \hat{n}_x^p \rangle \langle \hat{n}_{x+d}^p \rangle} - 1 \right), \tag{3}$$

in analogy with the hole–hole correlator $g_h^{(2)}$. We evaluate $g_{pair}^{(2)}$ on a subset of our data in which at least two pairs are present in the system. In the case of ladders containing four to five holes (Fig. 4a), we find a peak in the correlator at distance $|d| = 4$, whereas it peaks at distances $|d| = 2$ and $|d| = 4$ for the ladders containing six to seven holes. The pairs thus arrange in a spatial structure for which they maximize both their mutual distance and the distance to the edges of the system. This repulsion between pairs is an indication of the comparably high mobility of pairs in mixD settings, which could be further enhanced by increasing the leg tunnelling $t_\parallel$ in larger systems. Such highly mobile pairs can potentially reach very high critical temperatures[10] and can be an important ingredient for high-$T_C$ superconductivity.

To investigate the pair interactions numerically, we perform finite-temperature MPS simulations on a system with the same parameters as in the experiment ($L = 7$, $t_\parallel / J_\perp = 0.7$). Here pair interaction is directly revealed by the density of pairs $\langle \hat{n}_x^p \rangle$, which shows strong spatial dependence in the low-temperature regime (Fig. 4b) with an average distance of $|d| = 4$ ($|d| = 2$) for a system with four (six) holes.

The presence of sharp edges fixes the phase of the density modulation, thus leading to its visibility in the pair density as a direct consequence of our open boundary conditions and small system size. Such a modulation of the pair distribution is reminiscent of Friedel oscillations of indistinguishable fermions near an impurity[40] and of charge-density waves[41]. Larger systems are needed to distinguish between these effects.

In this work, we have demonstrated the direct observation of hole pairing in a quantum gas microscope setting. We have realized a paradigmatic model that reaches hole binding at high temperatures close to the spin-exchange energy and small pair size by engineering doped, mixD fermionic ladders. We confirm the effective mixD description, predicting that the suppression of Pauli repulsion enables the formation of a bound state.

This allows us to experimentally investigate the pairing mechanism based on hole motion and magnetism, thereby emphasizing the relevance of magnetic correlations as a potential origin of the charge carrier pairing underlying unconventional superconductivity. Furthermore, we have seen signs of significant mobility of the bound pairs through their repulsive interaction. Our approach with strongly bound and mobile pairs can lead to high critical temperatures and therefore can push the current temperature limits of high-$T_C$ superconductors[10]. Possible techniques to modify the interplay of kinetic and exchange energies in solid-state systems in the spirit of the mixD setting include changing lattice geometry and spin polarization[42], or controlling kinetic energy and the topology of electrons by varying the twist angle in moiré systems such as twisted graphene[43]. Our approach might also be realized in real materials as dynamical superconductivity using Floquet engineering to alter the effective exchange interactions[44]. Within the scope of quantum simulations, our technique can be readily extended to higher dimensions using bilayer quantum gas microscopes[38,45,46], and to even higher binding energies and pair mobility in larger systems at $t_\parallel > J_\perp$, or stripe formation at higher leg numbers[47]. Our results thus pave the way for the measurement of collective phases of bound pairs such as crystallization and superfluidity, and shed light on their competition[23,39,48,49].

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

## Methods

### Experimental sequence

In each experimental run, we prepare a cold atomic cloud of $^6$Li in a balanced mixture of the lowest two hyperfine states $|F = 1/2, m_F = \pm 1/2\rangle$. For evaporation, we confine the cloud in a single layer of a staggered optical superlattice along the $z$ direction with spacings $a_s = 3\,\mu m$ and $a_l = 6\,\mu m$, and depths $V_s = 51\,E_R^s$ and $V_l = 120\,E_R^l$, where $E_R^\alpha = h^2/(8Ma_\alpha^2)$ denotes the recoil energy of the respective lattices ($\alpha = s, l$), and $M$ is the mass of an atom. The atoms are harmonically confined in the $xy$ plane and the evaporation is performed by ramping up a magnetic gradient along the $y$ direction (see ref. [38]). The final atom number is adjusted by the evaporation parameters.

We adiabatically load the cloud into an optical lattice in the $xy$ plane with spacings $a_x = 1.18\,\mu m$ and $a_y = 1.15\,\mu m$. Simultaneously, we apply a repulsive potential using a DMD, which both compensates for the harmonic confinement of the Gaussian-shaped lattice beams and shapes the system into a geometry of four $2 \times 7$ ladders following the procedure described in ref. [37]. The DMD is also used to apply a spin-independent optical potential offset $\Delta$ between sites along the rung direction. The loading is performed in three steps (Extended Data Fig. 1). (1) A first stage, in which the two legs of each ladder are nearly disconnected, is reached by ramping the lattice depths to $V_x = 3\,E_R^x$ and $V_y = 20\,E_R^y$ in 100 ms. (2) The optical potential offset $\Delta$ (see Potential offset calibration) is applied to one leg of each ladder by instantaneously (<20 μs) switching the pattern of the DMD. (3) The lattice depths are ramped linearly to their final values $V_x = 12\,E_R^x$ and $V_y = 6\,E_R^y$ in 100 ms.

Interactions between the atoms are set by the $s$-wave scattering length $a_s$, which we adjust by applying a magnetic field close to the broad Feshbach resonance of $^6$Li around 830 G. The scattering length is increased from $a_s \approx 350a_B$ during evaporation, with $a_B$ being the Bohr radius, to $a_s \approx 1,310a_B$ in the final configuration. The resulting system is described by the Fermi–Hubbard model and an additional potential offset $\Delta$. Our parameters are the repulsive on-site interaction $U = h \times 4.29(10)\,kHz$, tunnelling $\tilde{t}_\parallel = h \times 78(10)\,Hz$ and $\tilde{t}_\perp = h \times 303(23)\,Hz$ and the offset $\Delta \approx 0.5U$ or $\Delta = 0$ depending on the configuration (mixD or standard). As $U/\tilde{t}_\perp$, $U/\tilde{t}_\parallel \geq 14$, the system can be effectively described by the $t$–$J$ model (see also 'From the Fermi–Hubbard to the $t$–$J$ model'). Note that we use $\tilde{t}_\perp$, $\tilde{t}_\parallel$ for the tunnelling parameters in the Hubbard model, and $t_\perp$, $t_\parallel$ in the $t$–$J$ model. Along the legs, the tunnel coupling is independent of $\Delta$ and is $t_\parallel = \tilde{t}_\parallel = h \times 78(10)\,Hz$, yielding a spin exchange of $J_\parallel = h \times 5.7(1.5)\,Hz$. Along the rungs, the mixD system ($\Delta/U \approx 0.5$) yields $t_\perp = 0$ and an enhanced spin exchange $J_\perp = h \times 114(42)\,Hz$. Without the potential offset, that is, in the standard system ($\Delta = 0$), tunnelling is unaffected, leading to $t_\perp = \tilde{t}_\perp = h \times 303(23)\,Hz$ and $J_\perp = h \times 86(13)\,Hz$.

### Potential offset calibration

We realize the mixD system by applying a local spin-independent light shift $\Delta$ to one of the legs on each ladder. The amplitude is directly proportional to the light intensity, which is controlled by the DMD. Calibration of the offset is performed by running the experimental sequence described above for different light intensities, and measuring the density of doubly occupied sites (doublons) in the system. An increase of doublons is seen when $\Delta \approx U$, that is, when the lowest band of the upper leg becomes resonant with the interaction band of the lowest leg (Extended Data Fig. 2). This calibration was repeated several times throughout data collection, with typical shift of the doublon peak of around 10%. We attribute these calibration differences to drifts in the beam shape of the light that is sent to and diffracted from the DMD, yielding an overall estimation of the uncertainty on $\Delta$ of about ±15%. Such uncertainty in $\Delta$ is not critical for realizing a mixD setting and mostly influences the value of $J_\perp$.

### Suppression of rung tunnelling

The potential offset $\Delta \gg \tilde{t}_\perp$ between the two legs shifts the energy levels between neighbouring sites and thus suppresses tunnelling along the rungs[47]. Doublon–hole pairs, however, become biased in the mixD system and appear predominantly as double occupancy on the leg with lower potential and corresponding empty site on the upper leg. Although in the standard system doublon–hole pairs along the rung appear with probability $\propto (\tilde{t}_\perp/U)^2$ (ref. [46]), the potential offset in the mixD case lowers the energy difference between the singly occupied state and the doublon–hole pair to $U - \Delta$. This effect can be seen in the density of the mixD system in Fig. 1c. In Extended Data Fig. 3, we plot the same data after removing ladders containing double occupancies. The density imbalance mostly disappears, indicating minimal tunnelling during preparation.

### Detection

The data presented in the main text originate from two types of measurement: (1) charge-resolved and (2) spin-charge-resolved measurements. In both cases, the detection procedure starts by ramping the $xy$ lattices to $43\,E_R^{xy}$ within 250 μs, effectively freezing the occupation configuration. In the case of spin-resolved measurements (2), a Stern–Gerlach sequence separates the two spin species into two neighbouring planes of the vertical superlattice, which are then separated by 21 μm from one another using the charge pumping technique described in ref. [38]. Finally, fluorescence images are taken using Raman sideband cooling in our dedicated pinning lattice with an imaging time of 1 s (ref. [50]). For a charge-only measurement (1), only one plane is populated by atoms, whereas in the case of a fully resolved measurement (2) two planes are populated by the two different spin species. The fluorescence light of the atoms is then collected through a high-resolution objective and imaged onto a camera. For a fully spin-resolved measurement (2), the fluorescence of both planes is collected simultaneously and imaged on the camera, allowing the reconstruction of the atomic distribution of both spins with a single exposure. A charge-only measurement only allows the reconstruction of the atomic configuration, without any spin information.

The imaging technique and the pumping procedure both impact our overall detection fidelity. The imaging fidelity, which takes into account atom losses and atom displacement during the imaging procedure, is estimated by comparing two consecutive fluorescence pictures of the same atomic distribution, and we obtain an average imaging fidelity $\mathcal{F}_{I,1} = 98.7(1)\%$ and $\mathcal{F}_{I,2} = 98.2(2)\%$ per atom for charge-only and full-spin-charge resolution, respectively. The pumping fidelity is estimated by comparing the average number of atoms detected after pumping to the average number of atoms before pumping, and we obtain an average pumping fidelity of $\mathcal{F}_P = 97.6(1)\%$, taking into account the slight discrepancy between $\mathcal{F}_I$ and $\mathcal{F}_{I,b}$. We deduce an overall detection fidelity of $\mathcal{F}_1 = \mathcal{F}_{I,1} = 98.7(1)\%$ and $\mathcal{F}_2 = \mathcal{F}_{I,2}\mathcal{F}_P = 95.8(1)\%$ in the case of charge-only and full-spin-charge resolution, respectively.

### Data statistics

We have taken approximately 19,000 experimental shots, iterating between mixD $\Delta \approx U/2$ and standard $\Delta = 0$. Here 61% of the shots have charge-only resolution and 39% have full spin and charge resolution.

The ladders are very sensitive to small drifts in the DMD pattern relative to the lattice sites. We thus keep track of the ladder potential by continuous automatic evaluation of the charge distribution and automatic feedback to the DMD pattern. If the average leg-to-leg occupation imbalance of standard ladders exceeds two holes, we dismiss the respective set of data due to the uncontrolled drift in the potential. For data analysis we then only take into account ladders without double occupancies and with a leg-to-leg occupation imbalance of maximally one hole. This leaves us with more than 24,000 individual ladders, about half of which contain between two and four holes (Extended Data Fig. 4a). Most ladders show a magnetization $|M^z| < 2$, with $M^z = \sum_i \hat{S}_i^z$ (Extended Data Fig. 4b). Figures and values given in the main text, unless otherwise mentioned, are filtered for two to four holes.

## Numerical simulations using DMRG

We numerically simulate the $t$–$J$ model, equation (1) in the main text, using MPS. For the mixD ($t_\perp = 0$) case, we set the parameters to $J_\parallel/J_\perp = 0.047$, $t_\parallel/J_\perp = 0.7$. In the standard ($t_\perp > 0$) case, the parameters are $J_\parallel/J_\perp = 0.06$, $t_\parallel/J_\perp = 0.9$ and $t_\perp/J_\perp = 3.57$. This corresponds to the $t$–$J$ model derived from a Fermi–Hubbard model with $U/t_\perp = 14.16$, $t_\parallel/t_\perp = 0.26$ and, in the mixD case, $\Delta/U = 0.5$. We use the TeNPy package[51,52] to perform the MPS simulations. To simulate systems at finite temperature, we use the purification method[53,54], in which the Hilbert space is enlarged by an auxiliary site $a(i)$ per physical site $i$. The finite-temperature state of the physical system is obtained by tracing out the auxiliary degrees of freedom. We start from an infinite-temperature state, in which the physical and auxiliary degrees of freedom on each site are maximally entangled. In particular, we implement an entangler Hamiltonian[55] to prepare the infinite-temperature state of the $t$–$J$ model. We work in the grand canonical ensemble and thus introduce a chemical potential $\mu$ to control the average number of holes in the system. Starting from the infinite-temperature state, we then use the $W^{II}$-time-evolution method[56] to perform imaginary time evolution up to the desired temperature. Depending on the system size, model (standard $t$–$J$ versus mixD or Fermi–Hubbard), doping and temperature (finite temperature versus ground state), we use a bond dimension between $\chi = 50$ and $\chi = 400$. For the finite-temperature calculations, we use an imaginary time step of $dt/J_\perp = 0.025$. We have carefully checked our results for convergence in the bond dimension and the size of the time step. We have benchmarked the MPS calculations by comparing with exact diagonalization for small system sizes and find the same results.

To directly compare with the experimental data, we sample snapshots from the MPS using the perfect sampling algorithm[57]. In the evaluation of the snapshots, we account for the experimental detection fidelity by randomly placing artificial holes in the MPS snapshots according to our detection fidelity. We then apply the same filters regarding hole number and occupation imbalance as for the experimental data and model the hole number distribution of the experimental data (Extended Data Fig. 4a) by weighting the snapshots accordingly.

For ground-state simulations, for example to obtain the binding energies, we use the DMRG algorithm and work in a fixed $S_z^{\text{tot}}$ and particle number sector.

## From the Fermi–Hubbard to the $t$–$J$ model

The Fermi–Hubbard model

$$\mathcal{H} = -\sum_{\langle i,j\rangle,\sigma} -\tilde{t}_{ij}\, (\hat{c}_{i,\sigma}^\dagger \hat{c}_{j,\sigma} + \text{h.c.}) + U\sum_i \hat{n}_{i,\uparrow}\hat{n}_{i,\downarrow}$$

contains a hopping term and (repulsive) on-site interaction. An additional potential offset $\Delta$ on one of the two legs leads to

$$\mathcal{H}_\Delta = \mathcal{H} + \Delta \sum_{i\in(x,y=B)} \hat{n}_i,$$

which cannot be generally reduced to an effective Hamiltonian with a tunnelling $t_\perp(\tilde{t}_\perp, \Delta)$, because in general the physics will depend both on the underlying Fermi–Hubbard tunnelling amplitude $\tilde{t}_\perp$ and the offset $\Delta$. An effective description only exists in the regime $\tilde{t}_\perp \ll \Delta \ll U$ (as well as for $U \ll \Delta$ and the trivial $\Delta = 0$), where $\tilde{t}_\perp$ is eliminated from the Hamiltonian by working in a time-dependent basis. We mention that, even in this regime, the effective model does not capture the full physics, but only holds for intermediate timescales for which the system is in a metastable state. For small tilts $|\Delta| \ll |\tilde{t}_\perp|$, no such metastability exists but instead the system directly equilibrates to a state in which more holes are in the upper leg. Such a system is not described by an effective Hamiltonian with mixed dimensionality, but by the full Hubbard model with $\Delta$ and $\tilde{t}_\perp$ terms.

In the limit of large interactions $U \gg t$, where $U/\tilde{t}$ needs to be large enough to be well into the Mott-insulating regime, double occupancies are suppressed. An expansion to leading order in $\tilde{t}/U$ yields several terms, including the $t$–$J$ model of equation (1) with $J = 4\tilde{t}^2/U$ and thus $t \gg J$. In addition, the expansion yields terms of the order of $t^2/U$ (ref. [58]) describing next-nearest-neighbour hopping by a (virtual) double occupancy, in analogy with the spin-exchange term.

For our mixD system the only term arising is approximately $t_\parallel^2/U \ll J_\perp$, $t_\parallel$, which is much smaller than the relevant energy scales in the system and can thus be omitted. For the standard system there are more possible combinations of processes, such as approximately $t_\parallel t_\perp/U$, which is much larger than the process including only $t_\parallel$, but the system is still dominated by $t_\perp$, $t_\parallel$ and $J_\perp$. In the parameter regime in which $t_\parallel \gg t_\perp$, however, this term becomes increasingly important such that the Fermi–Hubbard system can eventually not be approximated by the $t$–$J$ model of equation (1). This explains discrepancies found in the literature between binding energies calculated in $t$–$J$ ladders[25,34] and in Fermi–Hubbard ladders[59] in the same parameter regimes.

## Temperature estimation

We estimate the temperature of our system by comparing the measured rung spin correlations $C(0, 1)$ as defined in equation (2) to the values calculated from MPS snapshots (Extended Data Fig. 5a). We find that our average rung spin correlations of $C(0, 1) = -0.38(1)$ for two to four holes correspond to a temperature of $k_B T = 0.77(2)J_\perp$.

Our data are, however, not well described by a single spin correlation value, as we see variations both in time and across the four simultaneously realized ladders. The temperature estimation for the full dataset is therefore an average, and the data can contain features of both lower and higher temperatures. One reason for temperature variations are drifts in the apparatus on a timescale of days, affecting in particular the evaporation stage, which sets the global temperature. Another reason is the potential shaping, which distributes entropy between the four ladders and the surrounding bath. We thus attribute a temperature to each ladder (out of the four ladders we realize simultaneously) and each point in time by averaging the spin correlations of a time window of about ±12 h. The resulting spin correlation and temperature distribution are shown in Extended Data Fig. 5b.

## Correlation functions

Evaluating correlators in finite-sized systems with fixed particle number leads to finite-size offsets, due to self-correlation of the particles. For our purpose we have to distinguish two cases. For correlations between different legs, for example, the rung hole correlation $g_h^{(2)}(0, 1)$, self-correlation does not cause problems. In the mixD case holes cannot move from one leg to the other, such that finding a hole in leg $A$ does not influence the number of holes in leg $B$. In the standard case holes are mobile between the legs, but the focus of the analysis still lies on holes in opposite legs, because we select the data for low occupation imbalance. The correlations are thus not influenced by self-correlation. Correlations within the same leg are, however, strongly affected by finite-size offsets. We correct for these offsets using

$$g_h^{(2)}(d, 0) = \frac{1}{\mathcal{N}_d} \sum_{i-j=(d,0)} \left( \frac{\langle \hat{n}_i^h \hat{n}_j^h \rangle}{\langle \hat{n}_i^h \rangle \langle \hat{n}_j^h \rangle} \frac{N_l}{N_l - 1} - \frac{L}{L-1} \right),$$

where $N_l$ is the number of holes in the leg and $L$ is the length of the leg. The same offset correction is applied to the pair correlator of equation (3) in the main text. The offset correction applies a distance independent correction and thus affects the overall value, but not the shape of the curve.

## Binding energy

We estimate the binding energy of holes from the measured correlation $g_h^{(2)}(0, 1)$ of two holes on the same rung. To this end, we simplify the

mixD $t$–$J$ Hamiltonian of equation (1) by neglecting the two smallest energy scales $J_\parallel$, $t_\parallel$. This is partly justified by the fact that both are below the estimated temperature $T$ of the experiment.

As a result, the Hamiltonian completely decouples into individual rungs and we can exactly diagonalize the latter. Then, as detailed below, we perform a canonical calculation of the entire system, with exactly one hole on each of the two legs of length $L$. From the known temperature $T$ and the rung superexchange $J_\perp$ we obtain a direct relation between the binding energy $E_b$ and the rung-correlation function $g_h^{(2)}(0,1)$:

$$E_b = -\beta^{-1} \ln \left[ \frac{(1 + 3e^{-\beta J_\perp})\left(1 - \frac{g_h^{(2)}(0,1)}{L-1}\right)}{4(1 + g_h^{(2)}(0,1))} \right] \quad (4)$$

where $\beta = 1/(k_B T)$.

To use the measured correlation value given in Fig. 2a, we have to eliminate the density dependence of the $g_h^{(2)}$ correlator. Using the insights of Fig. 3b, we scale the hole correlator with the hole density $n_h$. Using the scaled correlator $g_h^{(2)} n_h$ and the above formula with the experimentally estimated values for $k_B T/J_\perp = 0.77(2)$, we obtain the estimate for the binding energy $E_b = 0.82(6)J_\perp$ stated in the main text. The error derives from the error on the experimental value and the error on the temperature estimation. If we use the measured hole correlation for exactly two holes in the system (Fig. 3b), as is used in the above derivation of (4), we obtain a binding energy of $E_b = 0.79(9)J_\perp$. Both calculations yield results in very good agreement with the theoretical prediction from DMRG at $L = 7$ of $E_b^{theo} = 0.81J_\perp$. We calculate the binding energy in large systems using DMRG and find that the value settles quickly to around $E_{b,\infty} = 0.78J_\perp$. For length $L = 40$ rungs we find $E_{b,40} = 0.7805 J_\perp$ and for $L = 80$ rungs we find $E_{b,80} = 0.7797 J_\perp$. This demonstrates that our system with its tightly bound pairs provides a good approximation to the physics in larger systems.

In the remainder of this section, we explain the simplified model used here in more detail and derive from it equation (4). As mentioned in the beginning, we neglect the smallest energy scales $t_\parallel$ and $J_\parallel$. The eigenstates of each decoupled rung therefore become the two-hole state $|hh\rangle$, the four spin-hole states $|sh, y, \sigma\rangle$ with leg index $y = 0, 1$ and spin index $\sigma = \uparrow, \downarrow$, the spin-singlet state $|S\rangle$ and the three spin-triplet states $|T, m\rangle$ with $m = -1, 0, 1$. The corresponding eigenenergies are $\epsilon_{hh} = V$, $\epsilon_{sh} = \epsilon_T = 0$ and $\epsilon_S = -J_\perp$. Note that we allowed for a variable energy $V$ of the hh state. For $t_\parallel = J_\parallel = 0$ we know that $V = 0$; however, for small but non-zero couplings $t_\parallel$, $J_\parallel$, a non-zero renormalization of $V \neq 0$ can be expected. The strength of $V$ can be calculated perturbatively[22], but we treat it as a free parameter here, which allows us to go beyond a perturbative analysis.

We start by defining the binding energy of the simplified model in the thermodynamic limit $L \to \infty$. To this end, we compare the ground-state energy of a system with two independent holes, $2(E_{1h} - E_{0h}) = 2J_\perp$, with the ground-state energy of a system with one pair of bound holes, $E_{2h} - E_{0h} = V + J_\perp$; both are measured relative to the undoped ground state, $E_{0h} = -L J_\perp$. The binding energy is then defined as

$$E_b = 2E_{1h} - E_{0h} - E_{2h} = J_\perp - V.$$

For $E_b > 0$ ($E_b < 0$) the two-hole ground state is paired (unpaired).

To derive equation (4) we perform a canonical calculation with exactly one hole per leg. The probability of finding both holes on the same rung anywhere in the system becomes $p_{hh} = L e^{-\beta E_{hh}} Z_S^{L-1}/Z$, where we defined the spin $Z_S = e^{-\beta E_S} + 3e^{-\beta E_T}$ and total partition functions

$Z = L e^{-\beta E_{hh}} Z_S^{L-1} + 4L(L-1)e^{-2\beta E_{sh}} Z_S^{L-2}$. By the definition of the $g^{(2)}$ function provided in the main text, we obtain the relation

$$g_h^{(2)}(0,1) = \frac{p_{hh}/L}{(1/L)^2} - 1 \quad (5)$$

in our model by assuming a homogeneous density of $\langle \hat{n}_i \rangle = 2/(2L)$ on each site and a homogeneous probability for an existing pair to occupy a specific rung of $1/L$. Thus $\langle \hat{n}_i^h \hat{n}_j^h \rangle = p_{hh}/L$ for fixed $(i,j)$ on one rung. There are thus $L$ identical terms in the sum of $g_h^{(2)}$ and one arrives at equation (5) by inserting $\mathcal{N}_{(0,1)} = L$. Simplifying this expression and solving for $E_b$ finally yields equation (4).

## Data availability

Source data are provided with this paper. The datasets generated and analysed during the current study are available from the corresponding author on reasonable request.

## Code availability

The code used for the analysis are available from the corresponding author on reasonable request.

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

**Acknowledgements** We thank T. Giamarchi for discussions. We thank A. Kantian for discussions and critical reading of the manuscript. This work was supported by the Max Planck Society (MPG), the European Union (FET-Flag 817482, PASQUANS), the Max Planck Harvard Research Center for Quantum Optics (MPHQ), the German Federal Ministry of Education and Research (BMBF grant agreement 13N15890, FermiQP) and Germany's Excellence Strategy (EXC-2111-390814868). T.C. acknowledges funding from the Alexander v. Humboldt Foundation. F.G. acknowledges funding from the European Research Council (ERC) under the European Union's Horizon 2020 research and innovation programme (grant agreement no. 948141) from ERC Starting Grant SimUcQuam. A.B. acknowledges funding from the National Science Foundation (NSF) through a grant for the Institute for Theoretical Atomic, Molecular, and Optical Physics at Harvard University and the Smithsonian Astrophysical Observatory. E.D. acknowledges funding from the Army Research Office (ARO) (grant number W911NF-20-1-0163) and AFOSR-MURI: Photonic Quantum Matter award FA95501610323.

**Author contributions** S.H. led the project. S.H., T.C., P.B. and D.B. contributed significantly to data collection and analysis. A.B. carried out the numeric calculations. S.H., T.C. and T.A.H. wrote the manuscript. E.D., F.G., I.B. and T.A.H supervised the study. All authors contributed extensively to interpretation of the data and production of the manuscript.

**Funding** Open access funding provided by Max Planck Society.

**Competing interests** The authors declare no competing interests.

**Additional information**
**Correspondence and requests for materials** should be addressed to Sarah Hirthe.
**Peer review information** *Nature* thanks Randall Hulet for their contribution to the peer review of this paper. Peer reviewer reports are available.

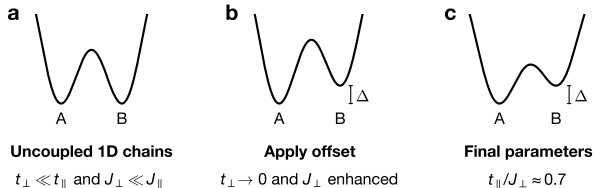

**a** Uncoupled 1D chains
$t_\perp \ll t_\parallel$ and $J_\perp \ll J_\parallel$

**b** Apply offset
$t_\perp \to 0$ and $J_\perp$ enhanced

**c** Final parameters
$t_\parallel / J_\perp \approx 0.7$

**Extended Data Fig. 1 | Preparation sequence for mixD systems. a**, We first prepare nearly uncoupled 1D chains in which the leg tunnelling exceeds the rung coupling. **b**, While the legs are decoupled, we apply the offset Δ to one leg of the ladder. **c**, The final parameters are reached by ramping down the leg coupling and ramping up the rung coupling. There, the potential offset Δ between legs prevents tunnelling from one leg to the other. Note that in the final configuration $J_\perp \gg J_\parallel$.

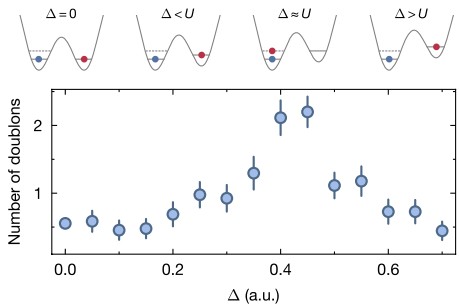

**Extended Data Fig. 2 | Calibration of the optical potential offset.**
The experimental sequence is run for different value of Δ in a regime close to
unit occupancy of the lattice. tunnelling from one leg to the other is suppressed
as long as $|\Delta - U| > 0$. When $\Delta - U$, tunnelling is possible, and an increased number
of doublons in the system is measured.

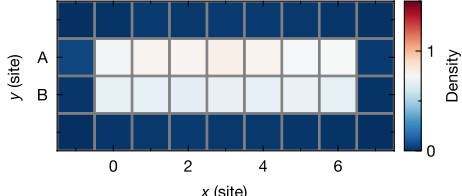

**Extended Data Fig. 3 | Density of the mixD system without doublons.**
The density of the mixD system, where only ladders without double occupancies
are taken into account.

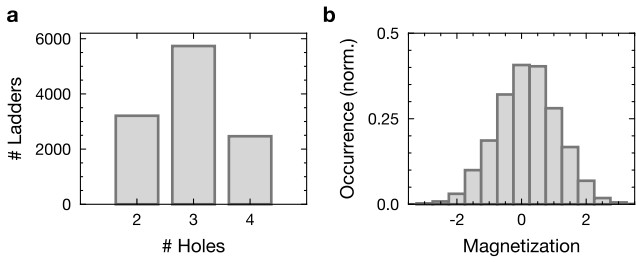

**Extended Data Fig. 4 | Hole and magnetization statistics. a,b**, Experimental distribution of holes per ladder (**a**) and total magnetization (**b**) for the data shown in Fig. 2a–c, and Fig. 3a.

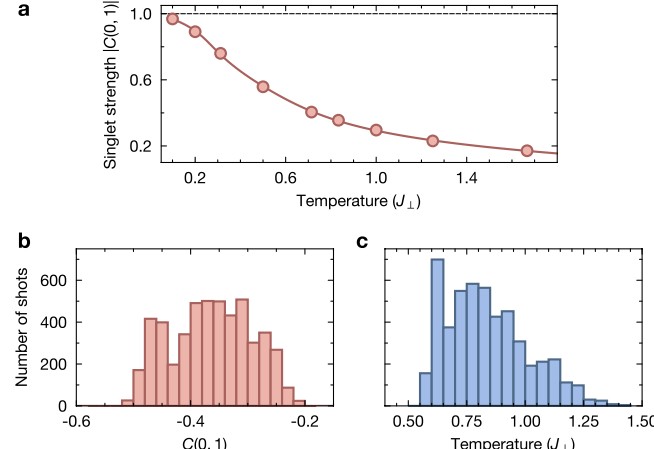

**b**

**c**

**Extended Data Fig. 5 | Temperature estimation. a**, Singlet strength versus temperature. The calibration of temperature is performed using MPS data containing two to four holes. **b**, Experimental singlet strength and **c**, inferred temperature distributions. We evaluate our rung spin correlations $C(0, 1)$ on the mixD system, using a time window of about 24 h. The temperature is extracted from $C(0, 1)$ using the MPS simulation (**a**).