## [Peer Review File · Nature]

Manuscript Title: Magnetically mediated hole pairing in fermionic ladders of ultracold atoms

Reviewer Comments & Author Rebuttals

Reviewer Reports on the Initial Version:

Referees' comments:

Referee #1 (Remarks to the Author):

The authors observe hole pairing due to magnetic correlations in the gas of Li6 loaded into optical lattice that takes a shape of a two leg ladder with seven rungs. The authors report a hole-hole binding energy to be comparable to the superexchange interaction and claim the repulsion between bound hole pairs in their observation of the spatial structures in the pair distribution.

Overall I think the manuscript is interesting for general audience, clearly written and can bring some new insights into understanding the generic pairing mechanism.

The weakness of the manuscript is very small system size, thus the conclusion made by the authors in the worst case can turn out to be a result of a finite-size effect. I understand that it might be challenging to realize longer ladders experimentally (the authors, however, could be more specific on what limits their experiments), but one can definitely perform numerical simulations to show that the picture and the conclusion remain the same for much longer chains

I cannot recommend the manuscript for publication in Nature before the conclusion is tested (or at least analyzed) against the possible finite-size effects.

In addition, I have a few remarks regarding the numerical method:

- DMRG has been used. Given that the total size of the system is only 14 sites and there are two conserved U(1) symmetries – total magnetization and particle number, can one simulate the system with exact diagonalization? What is the advantage of MPS in this case?
- Details of DMRG/finite temperature algorithm that control the accuracy are not specified: please include either truncation error or bond dimension?
- around line 850 "In the limit of large systems the DMRG gives a binding energy of $E_b \propto 0.78 J_{\perp}$, demonstrating that our system with its tightly bound pairs provides a good approximation to the physics in larger systems.."
What "large systems" mean? Where one can see the data supporting this statement (Fig/Ref)?

Spotted typos:

- line 75 "also also"
- Figure S5: "b," should be "b,c". I think, one has to show error-bars in this Figure.
- Eq. (5) should it be "ln" instead of "log"?

In the manuscript "Magnetically mediated hole pairing in fermionic ladders of ultracold atoms" by S. Hirthe et al., the authors reported an impressive work on a quantum simulation of cold-atom Fermi-Hubbard or t-J Hamiltonian system. In this work, by utilizing a quantum gas microscopy, the group successfully observed the pairing of holes in a doped ladder system. The background of this work is closely related with the unconventional superconductivity where the pairing of carriers of holes or electrons through magnetic fluctuations in a doped magnet is believed to be responsible for the superconductivity. To get deeper understanding of mechanisms of high-Tc superconductor is one of the most important targets of a cold-atom quantum simulator. So, the present work certainly takes an essential step in the right direction for this field, although the lattice configuration the authors studied is a ladder system where the hole binding or correlation is artificially enhanced, and reliable theoretical calculations are available, different from a uniform 2D square lattice usually considered as a model lattice system for high-Tc superconductor. It is true, on the other hand, that such a flexibility of designing a system is one of great advantages of cold-atom quantum simulators. In particular, by realizing mixed-dimensional ladders, the authors suppressed the Pauli repulsion and enhanced the binding between holes along the rung, leading to the successful observation of the hole pairing. The authors made a systematic comparison between the cases of mixed dimensional and standard ladder lattices. In the supplementary information, the authors presented the results of the measurement of the spin-spin correlation around a hole pair, as well as the theoretical calculations supporting the arguments discussed in the main text, such as hole-hole correlation strength for different binding energies and temperatures, even-odd-number behaviors of the spin-spin correlation, hole-hole correlations both in mixed dimensional and standard ladders in a larger system, and pair distributions for difference sizes and coupling strengths.

The authors presented both experimental and theoretical results of the systematic investigations of the hole pairing. In particular, they evaluated various quantities such as hole-hole correlations between and within the legs, and the spin-spin correlations between the legs. In addition, the temperature and hole-number dependences of the hole-hole correlation is investigated, where the atomic temperature in the ladder is estimated from the spin-spin correlation. Furthermore, the distribution of the hole pairs are studied for different hole numbers, and compared with the theoretical calculations.

The presented result is quite impressive, and the data are reliable enough, so I would like to recommend the paper for publication for Nature. Below, I point out minor issues, which the authors could consider to further improve the readability of the paper.

L43) Although the authors mentioned in the abstract, "we delineate a novel strategy to increase the critical temperature for superconductivity", this seems somewhat strong. I expected sound arguments but could only find some related arguments on the possible cold-atom realizations in the last part of the main text. It is nice if they could add relevant arguments or weaken the words otherwise.

L107) It is helpful if the authors give an explicit expression of t_{\perp} as functions of the potential offset Δ , and Hubbard parameters, probably in L153 or in "From the Fermi-Hubbard to the t - J model" in Methods.

Fig. 2(a)) Is there any theory calculation for the data of the standard ladder? Any effect by the additional holes also in the standard ladder ?

Fig. 2(c)) Are there the corresponding data for the case of the standard ladder ? Since this quantity is that within the leg, a similar behavior would be expected, but if the authors have the data, it would be better to show them.

Fig. 2 (d)) Why is the correlation of the standard ladder larger than that for the mixed D in the absence of hole($N_h=0$) ?

Fig.3(a)) For the standard ladder, the two data points at smaller singlet strengths show smaller hole-hole correlations than other data, in spite of the statement of "remain mostly constant" in L299. Is any qualitative explanation available? Again, did the authors perform any theoretical calculations for the standard ladder ?

L263) The alternating behavior is not observed for larger hole numbers in Fig. 2d). Actually, the correlation takes its minimum at $N_h=4$, contrary to the expectation. Any reason?

Fig.4(a) and (b)) What is the atomic temperature in this measurement? The lowest temperature is about $0.6 J_{\perp}$ in the data of Fig. 3(a), at which the theoretical calculations in Fig. 4 (b) do not show apparent distributions of the hole pairs. This looks inconsistent with the experimental results of Fig. 4(a).

Fig. S6) How do you understand the behavior of the smallest binding energy of $0.3 J_{\perp}$? Why does the correlation take a negative maximum at a certain temperature?

L884) It would be helpful to add some more detail about the derivation of the relation Eq.(7).

Can you comment on the possibility of experimental study of higher filling cases, instead of hole-doping? Such a case is closely related with the e-doping high- T_c superconductivity.

Referee #3 (Remarks to the Author):

The authors report on quantum simulation of hole pairing in a two-leg ladder of lattice sites containing pseudo-spin-1/2 fermionic atoms. The system consists of 7 rungs on a ladder on which the experimenters could locally address single sites, and apply an off-setting potential along the rungs of the ladder so as to control the perpendicular tunneling. Without the potential, the system realizes the standard Hubbard model, which has been studied previously by the Munich group and others. With the potential, the perpendicular spin-exchange rate can be made much larger than the parallel one, thus producing strong singlet pairing along the rungs. This mixed dimensional configuration is dubbed mixD. As explained theoretically in Ref. 10 (by a subset of the present authors), the mixD configuration can suppress Pauli blocking of holes on short length scales. The hole-pair binding energy is predicted to increase, thus significantly increasing hole pairing of nearest neighbors.

This understanding is supported by measurements of the hole-hole correlation function, which is peaked for holes on the same wrung in the mixD configuration, but not for the standard one for which the correlations are negative for holes on the same wrung.

I consider this work to be at the pinnacle, thus far, of quantum simulation with ultracold atoms, and particularly, with the quantum gas microscope. But the role of the finite size of the simulation in these experiments is also an important question. In the present case, the system size is only 2×7 , and as shown in Fig. 4, the system is strongly affected by finite size effects. How might these correlation functions be modified in the thermodynamic limit? Or in the limit of a 2D sheet? Some speculation along these lines could be valuable.

Also, the authors should be explicit about temperature limitations in these experiments, which are evident in Fig. 3, which shows that the maximum value of $g(2)$ is considerably smaller in magnitude than in the low temperature limit.

Finally, it is relevant for an article in Nature to add perspective, especially about the implications for real superconductors. The authors could compare the time it took to calculate the correlation functions with DMRG to achieve similar accuracy as the experiment, and to discuss how the insight

gained from quantum simulation complements direct calculation, in this case.

The authors have advanced new methods to perform this remarkable experiment. Furthermore, this experiment is relevant to one of the most vexing problems in strongly correlated many-body systems. With suitable responses, I strongly recommend publication in Nature.

Response to the Referees

Magnetically mediated hole pairing in fermionic ladders of ultracold atoms

Hirthe *et al.*

August 1, 2022

Introductory statement

We thank the referees for their careful reading as well as their positive and helpful feedback. Their comments were very valuable to us and their questions sparked interesting discussions with new insights. The referees spotted several points that profit from further clarification. This has led to several changes in the manuscript such as a more detailed discussion of finite size and finite temperature effects, giving more perspective regarding real superconductors and an additional section in the supplemental material, where we discuss the experimental and theoretical data for the standard ladder. We have also slightly reduced the length of the manuscript in order to stay within the length specifications for the main text.

Changes to the Manuscript

- Emphasising of finite size and finite temperature effects
- Emphasising the relevance of our work regarding higher temperature superconductivity
- Rewriting of the conclusion to include more perspective on real superconductors
- Shortening and reformulation in order to fulfil length requirement

Changes to the Methods

- Added details about the accuracy of the DMRG/MPS
- Added an explanation of the effective tunneling t_{\perp}
- Added explanations of the calculation of the large system size binding energy
- Added explanation of Eq. (7)

Changes to the Supplementary Material

- Added explanation of the theoretically predicted negative correlation for $g_h^{(2)}(0, 1)$ for $E_b = 0.3 J_\perp$
- Added a figure (Fig. S7) containing the theory curve and more experimental data for the standard ladder and added a corresponding section
- Added a figure (Fig. S10) and section discussing finite size effects in the hole correlation
- Added a subfigure (in Fig. S12) showing the pair correlations for a large system and discussing finite size effects

Figure R1: **Numerical hole-hole correlation in different system sizes.** **a**, Hole-hole correlation $g_h^{(2)}(|d|, 1)$ in a mixD system of different lengths between $L = 7$ and $L = 40$ at a temperature of $k_B T = 0.7 J_\perp$ calculated using MPS. The hole doping is kept constant at 15 % to 30 %. The points at $|d| = 0$ are plotted against system size in **b**.

Reply to Referee 1

We thank the referee for their constructive feedback. Their comments have helped to clarify the most relevant issues that were left unclear in our original manuscript. We are also grateful to the referee for spotting typos and imprecise notation.

- (i) The weakness of the manuscript is very small system size, thus the conclusion made by the authors in the worst case can turn out to be a result of a finite-size effect. I understand that it might be challenging to realise longer ladders experimentally (the authors, however, could be more specific on what limits their experiments), but one can definitely perform numerical simulations to show that the picture and the conclusion remain the same for much longer chains

The referee points out correctly that one of the main limitations of our experimental apparatus is the small system size. This is mostly due to the strong harmonic confinement of our optical lattice. We use a digital micromirror device (DMD) with light at 650 nm to generate a repulsive potential on top of the optical lattice. This repulsive potential serves three purposes. It compensates for the harmonic confinement, it shapes the ladder potential and it generates the potential offset Δ . The system sizes we realise are thus limited by the available incoherent light power sent to the DMD. This is however not a fundamental limit and will be overcome in future experiments.

The experimentally realised system of length $L = 7$ is however still much larger than than the size of hole pairs, which are strongly bound on the same rung. They are thus not strongly affected by the system size. Figure R1, which we have also added to the supplemental material, shows the numerical hole-hole correlator for different system sizes at our temperature. The shape of the curve is mostly independent of system size and the main effect of small system sizes is a negative finite size offset for distances $|d| > 0$. For the observation of pair interaction (Fig. 4 of the manuscript), the finite size of the system can influence the correlations as discussed in the text: the edges of the system pin the position of the pairs and thus the pattern. In a larger system the same number of holes would lead to a different pattern, because the pairs can have a larger distance to each other.

- (ii) DMRG has been used. Given that the total size of the system is only 14 sites and there are two conserved U(1) symmetries – total magnetisation and particle number, can one simulate the system with exact diagonalization? What is the advantage of MPS in this case?

Exact diagonalization is in principle possible for the 7×2 system, but especially for finite temperature the calculations are time-intensive. The advantage of MPS however is not only the shorter run-time, but mostly the possibility to compare to larger systems using the same method. We thus decided to use this method consistently throughout the paper. For small system sizes we benchmarked the MPS calculations by comparing to exact diagonalization and find the same results.

- (iii) Details of DMRG/finite temperature algorithm that control the accuracy are not specified: please include either truncation error or bond dimension?

Depending on the system size, model ($t - J$ standard versus mixed dimensional or Fermi-Hubbard), doping, and temperature (finite temperature versus ground state), we use a bond dimension between $\chi = 50$ and $\chi = 400$. For the finite temperature calculations, we use an imaginary time step of $dt/J_{\perp} = 0.025$. We have carefully checked our results for convergence in the bond dimension as well as the size of the time step. We have now included these details in the supplementary material.

- (iv) around line 850 “In the limit of large systems the DMRG gives a binding energy of $E_{b,\infty} = 0.78J_{\perp}$, demonstrating that our system with its tightly bound pairs provides a good approximation to the physics in larger systems..”

What does “large systems” mean? Where one can see the data supporting this statement (Fig/Ref)?

We thank the referee for pointing out this unclear statement of ours. We have compared the binding energy calculated using DMRG for different system lengths and find that the value settles quickly to around $E_{b,\infty} = 0.78 J_{\perp}$. For length $L = 40$ rungs we find $E_{b,40} = 0.7805 J_{\perp}$ and for $L = 80$ rungs we find $E_{b,80} = 0.7797 J_{\perp}$.

- (v) line 75 “also also”

We have corrected this.

- (vi) Figure S5: “b,” should be “b,c”. I think, one has to show error-bars in this Figure.

We changed the figure labels. Figure S5a has errorbars which are smaller than the markers. We now mention this in the figure caption. Figure S5b and c do not have errorbars in y -direction, because we intend to show the actual temperature distribution of our data set. We do not think errorbars would be meaningful on this quantity. Figure S5b and c do not show errorbars in x -direction either. Here we agree with the referee that errorbars are meaningful. However, the errorbars on our average spin correlations are considerably smaller than the width of the binning. We thus did not put errorbars in x -direction. We now mention this in the figure caption.

- (vii) Eq. (5) should it be “ln” instead of “log”?

We changed the notation to avoid confusion.

Reply to Referee 2

We are grateful to the referee for their constructive feedback and several excellent questions. Amongst others, they noted a limited discussion of the standard ladder results. We want to concentrate the discussion in the manuscript to the mixD results, as the emergence of pairing in this setting is the main result of our work. The standard ladder merely acts as a comparison to illustrate the effect of mixed-dimensionality. We however agree with the referee that the data and theory comparison is of interest to some readers and should be available. We thus performed additional calculations for that setting and added a section in the supplemental material.

The referee has furthermore brought up very interesting questions regarding specific data points and their interpretation, which we are discussing here in detail. Given the length restriction, for which we already needed to cut the manuscript, we could not include all of these discussions in the main text, but added most of the points to the supplementary material.

- (i) L43) Although the authors mentioned in the abstract, “we delineate a novel strategy to increase the critical temperature for superconductivity”, this seems somewhat strong. I expected sound arguments but could only find some related arguments on the possible cold-atom realisations in the last part of the main text. It is nice if they could add relevant arguments or weaken the words otherwise.

Even though we have only demonstrated pairing, and not superconductivity, we would like to stress the connection between the two. Pairing is a prerequisite for superconductivity and thus sets an upper bound for the critical temperature. Additionally, the pairs in such mixD systems display a high mobility (see Fig. 4 of the Manuscript and Fig. S8 of the SI), which can be increased even further by increasing t_{\parallel} . This high mobility in the mixD system, together with the extremely high binding energy, can potentially lead to very high critical temperatures [1]. We have thus experimentally shown a new route that not only boosts the binding of dopants, but also has the potential to increase the temperature for superconductivity. For our findings to have implications for real superconductors, one needs to find ways to modify the interplay of kinetic and exchange energies in solid state systems. One way to suppress the kinetic energy without necessarily suppressing the super-exchange interaction might be achieved by changing the lattice geometry and spin polarization. For example, in a weakly doped system of fermions on a triangular lattice in the regime of strong spin polarization, the kinetic energy of holes is geometrically frustrated and hence strongly suppressed [2]. In moiré systems, such as twisted graphene or TMDC materials, one often finds systems that include a combination of flat and broad bands. One can control kinetic energy and topology of electrons in the narrow band by varying the twist angle. On the other hand, magnetism of electrons in the narrow band may be dominated by interaction with electrons in the wider bands [3]. Furthermore, there are already reports of materials like the ladder compound $\text{Sr}_{14}\text{Cu}_{24}\text{O}_{41}$, that display effects that resemble the results in our mixD system [4, 5]. These might be explained by very similar underlying physics, and in this case already constitute an example of a material realization of our approach.

Our mixD setting might also be realised in an approach more closely related to our experimental technique, which probes pairing in a metastable state. One example is to use Floquet engineering to alter the effective exchange interactions in magnetic materials, as suggested, for example, in [6]. We thereby also open new regimes for dynamical superconductivity. We

now mention these possibilities in the conclusion of the manuscript.

- (ii) L107) It is helpful if the authors give an explicit expression of t_{\perp} as functions of the potential offset Δ , and Hubbard parameters, probably in L153 or in “From the Fermi-Hubbard to the t - J model” in Methods.

We thank the referee for pointing out that the explanation for the effective t_{\perp} is not clear. We can not give a general formula for a reduced effective t_{\perp} , because in general the physics will depend both on the underlying Fermi-Hubbard tunneling amplitude \tilde{t}_{\perp} and the offset Δ . An effective description only exists in the regime $\tilde{t}_{\perp} \ll \Delta \ll U$ (as well as for $U \ll \Delta$ and the trivial $\Delta = 0$), where \tilde{t}_{\perp} is eliminated from the Hamiltonian by working in a time-dependent basis. We would like to mention that even in this regime, the effective model does not capture the full physics, but only holds for intermediate timescales where the system is in a metastable state. For small tilts $|\Delta| \ll |\tilde{t}_{\perp}|$ no such metastability exists but instead the system directly equilibrates to a state where more holes are in the upper leg. Such a system is not described by an effective Hamiltonian with mixed dimensionality, but by the full Hubbard model with Δ and \tilde{t}_{\perp} terms. We have added a description in the Methods section as the referee has suggested.

- (iii) Fig. 2(a)) Is there any theory calculation for the data of the standard ladder? Any effect by the additional holes also in the standard ladder ?

Figure R2a, which we have also added in the supplemental material, shows the theory prediction of the hole-hole correlator for the standard ladder as well as the experimental data. Both show strong repulsion of holes on the same rung, although the experimental signal is weaker in amplitude than the numerical simulations. We discuss below potential reasons for such a discrepancy and why the mixD ladders are less affected than the standard ladders.

The weaker repulsion that we observe in the experiment compared to theory is, most likely, attributed to inhomogeneities of the scalar potential engineered by our DMD. Even though we aim at creating a flat potential, we can only reach a finite resolution, while factors like residual light speckles or relative position of the DMD potential with respect to the optical lattice change in time, and thus can not be detected in the time-averaged and site-resolved density. Drifts as small as a fraction of a lattice site can indeed already create a potential offset on the order of t_{\perp} , and MPS simulations show that a disorder of order t_{\perp} on the standard ladder decreases hole repulsion by a factor 3, which is already a larger suppression than our observations. In the case of mixD ladders, however, the dynamics along the rungs are already quenched by a large potential offset $\Delta \sim U/2$ between the legs. Small fluctuations of the potential do not affect the quenched tunneling and hardly affect the spin superexchange — the corrections are in fact well within the errorbars of our system parameters — and thus have a negligible impact on our observations.

The referee is correct that additional holes affect the hole correlation at larger distances in both the mixD and the standard ladders. In the standard case, this can be seen by the two weak maxima at distances $d = 2$ and $d = 5$ in figure R2a. Such a behaviour is consistent with mutual repulsion of the holes, forming an alternating arrangement on the ladder. As expected, this is different to the mixD ladder, where we can explain the pattern by a first hole on the other leg being attracted, and a second hole being repelled.

Figure R2: **Hole-hole correlation in the standard ladder.** **a**, Hole-hole correlation $g_h^{(2)}(|d|, 1)$ in the standard ladder, as shown in Fig. 2 of the main text, compared to the theoretical prediction at $k_B T = 0.7 J_\perp$ calculated using MPS (shaded line). **b**, Hole-hole correlation $g_h^{(2)}(|d|, 0)$ of holes in the same leg for the standard ladder with the theoretical prediction at $k_B T = 0.7 J_\perp$ calculated using MPS (shaded line).

- (iv) Fig. 2(c) Are there the corresponding data for the case of the standard ladder ? Since this quantity is that within the leg, a similar behaviour would be expected, but if the authors have the data, it would be better to show them.

The repulsion of holes within the same leg can be seen in Fig. R2b, which we have also added to the SI. The repulsion of holes is visible, but has a higher noise and smaller amplitude than for the mixD data. In the standard ladder the position of holes in a single leg is highly influenced by the number and position of holes in the other leg, whereas in the mixD system the paired holes do not influence the repulsion within a leg.

- (v) Fig. 2 (d) Why is the correlation of the standard ladder larger than that for the mixed D in the absence of hole ($N_h = 0$) ?

Without holes, the two systems are very similar and the equilibrium spin correlations directly map to temperature. We assume that the application of the tilt slightly heats the mixD system resulting in a lower correlation strength. The much stronger effect of hole doping on spin correlations (slope of the linear fit on Fig. 2d) in the standard case however, is unaffected by a small difference in temperature.

- (vi) Fig.3(a) For the standard ladder, the two data points at smaller singlet strengths show smaller hole-hole correlations than other data, in spite of the statement of “remain mostly constant” in L299. Is any qualitative explanation available? Again, did the authors perform any theoretical calculations for the standard ladder ?

It is correct that the data points with hottest temperature show a lower correlation than the rest of the points. They however also show comparatively large errorbars and their values are thus consistent with other points of the curve. These points are considerably hotter than our usual temperature and consequently have rather low statistics. The hotter temperature might also be caused by preparation irregularities which not only affect the temperature but also affect the physics, for example through the system flatness. Except for their unusual temperature and low statistics we do not have further reason to dismiss these points and thus included them in the figure. Theoretical calculations show a negligible effect of temperature on these correlations, which is in qualitative agreement with our experimental data. As

Figure R3: **Theoretical pair density.** Pair density calculated using MPS for the experimental system parameters. The data is taken from Fig. 4 of the manuscript. The colourbar is adjusted to highlight the pattern at a temperature of $0.6 J_{\perp}$.

can be seen in Fig. R2, the theory prediction agrees qualitatively but not fully quantitatively with the data.

- (vii) L263) The alternating behaviour is not observed for larger hole numbers in Fig. 2d). Actually, the correlation takes its minimum at $N_h = 4$, contrary to the expectation. Any reason?

The referee correctly points out that the data point at $N_h = 4$ behaves contrary to the naive expectation from hole pairing. We do not have an explanation based on the Fermi-Hubbard model, but suspect experimental imperfections to be the cause of this behaviour. We do expect the alternating pattern to wash out for larger N_h as long as the pairing probability is not unity. The data points at $N_h = 4, 5, 6$ are all comparable in strength.

- (viii) Fig.4(a) and (b)) What is the atomic temperature in this measurement? The lowest temperature is about $0.6 J_{\perp}$ in the data of Fig. 3(a), at which the theoretical calculations in Fig. 4 (b) do not show apparent distributions of the hole pairs. This looks inconsistent with the experimental results of Fig. 4(a).

The data of Fig. 4 is taken from the same measurement as the data for the other figures, for which we estimate a temperature of $0.77 J_{\perp}$. We evaluated the temperature of the subset that contributes to the figure (hole number and occurrence of pairs) by comparing to DMRG snapshots with the same selection criteria and find a temperature consistent with the aforementioned one. The referee is correct that this is a temperature average and our data is composed of subsets of varying temperature, with the coldest measurements being at about $0.6 J_{\perp}$. In Fig. R3 we show the theoretical calculations of Fig. 4b of the manuscript, but with an adjusted colourmap. It becomes clear, that there still is a nontrivial distribution of hole pairs. Since all data points, including the ones in Fig. 4, will be provided with the publication, we did not add this figure to the supplementary material.

- (ix) Fig. S6) How do you understand the behavior of the smallest binding energy of $0.3 J_{\perp}$? Why does the correlation take a negative maximum at a certain temperature?

We thank the referee for this interesting question. The behaviour for small binding energies also puzzled us when we saw it for the first time. The situation of negative $g_h^{(2)}$ at intermediate temperatures happens when the binding energy is much smaller than the spin gap of the system. The reason is, that the spectrum features more low-lying unbound states, than low-lying bound states, despite the ground state being a bound one.

Let us consider only states that do not break the singlet gap, that is the low-lying states. A ladder with a pair now has all spin directions fixed, because they all form singlets. In a ladder with an unbound pair, however, there are four different combinations of spin states, that do not break the singlet gap. The unbound states are E_b higher in energy than the paired state, but for intermediate temperatures $T \gtrsim E_b$, due to the minimisation of free energy, the system will be more likely to occupy an unbound state than a paired state.

- (x) L884 It would be helpful to add some more detail about the derivation of the relation Eq. (7).

The equation is derived assuming a homogeneous density of $\langle \hat{n}_i \rangle = 2/(2L)$ on each site and a homogeneous probability for an existing pair to occupy a specific rung of $1/L$. Thus $\langle \hat{n}_i^h \hat{n}_j^h \rangle = p_{hh}/L$ for fixed (i, j) on one rung. There are thus L identical terms in the sum of $g_h^{(2)}$ and one gets Eq. (7) inserting the above and $\mathcal{N}_{(0,1)} = L$. We have added this explanation to the supplemental material.

- (xi) Can you comment on the possibility of experimental study of higher filling cases, instead of hole-doping? Such a case is closely related with the e-doping high- \$T_C\$ superconductivity.

Within the framework of the plain Fermi-Hubbard model without next-nearest neighbour tunneling, hole doping and doublon doping are equivalent, whereas in high- T_C superconductors the doping level alters the Hamiltonian parameters and there are non-zero diagonal hopping elements. Electron doping is thus not equivalent to hole doping in those materials. For technical reasons mostly involving the barrier height of our DMD shaped potential, we chose to hole dope the experimental system. We however expect to find similar signatures if we conducted the experiment with doublon doping.

Figure R4: **Pair density in a large system.** The pair density n_{pair} in the ground state is plotted for a system of length $L = 50$ and a doping of 20 %. The shaded area corresponds to one s.e.m.. The data is calculated using DMRG.

Reply to Referee 3

We thank the referee for their constructive feedback and positive evaluation of our work. The three points brought up by the referee are all highly relevant and helped improve the manuscript. Especially the additional interpretation of the data and its implications for real superconductors will help a general audience to understand the relevance of our work.

1. [..]. But the role of the finite size of the simulation in these experiments is also an important question. In the present case, the system size is only 2×7 , and as shown in Fig. 4, the system is strongly affected by finite size effects. How might these correlation functions be modified in the thermodynamic limit? Or in the limit of a 2D sheet? Some speculation along these lines could be valuable.

The referee correctly noted that finite size effects are most relevant in Fig. 4. The correlations are affected by the small system size, in particular due to the sharp walls at the edges of the system and the short relative distance of both edges to each other. Fig. R4 shows the pair density in the ground state of a system with size $L = 50$ and 20 % doping. The density pattern is most pronounced at the edge of the system. After an initial decay the amplitude stays rather constant within the bulk of the system. We believe that this behaviour is representative for the thermodynamic limit, where we expect a charge-density-wave to form. We have added Fig. R4 as a subfigure of Fig. S11 to the supplementary material. Whether the behaviour is similar for a 2D system is a highly interesting question. In a 2D plane of mixed-dimensions we do not expect the formation of bound hole pairs, but we expect stripes to form along the direction of suppressed hopping. The nature of possible correlations between several stripes in the mixD setting is currently a matter of investigation. In a mixD bilayer setting on the other hand, strong pairing is expected, similar to the mixD ladder system. Numerical simulations for bilayer cylinder of up to 4 legs show strong binding. For larger 2D bilayer systems, which are not accessible with numerical methods anymore, strong binding is predicted by an effective theory [1]. We aim to realise this scenario experimentally in a future project.

2. Also, the authors should be explicit about temperature limitations in these experiments, which are evident in Fig. 3, which shows that the maximum value of $g(2)$ is considerably smaller in magnitude than in the low temperature limit.

We are grateful for the opportunity to address this point. The experiment is working in a temperature regime which is very similar to the binding energy and we thus do not expect a large ground state fraction. The achievement of this work instead lies in pushing the binding energy to a magnitude where binding can be observed at all. Our hole correlations reach a maximum value of $g_h^{(2)} = 0.3 \pm 0.1$ for our coldest temperatures, whereas values of $g_h^{(2)} > 1.2$ are expected for very low temperatures. We now clearly state the distance to the ground state in the main part of the manuscript.

3. Finally, it is relevant for an article in Nature to add perspective, especially about the implications for real superconductors. The authors could compare the time it took to calculate the correlation functions with DMRG to achieve similar accuracy as the experiment, and to discuss how the insight gained from quantum simulation complements direct calculation, in this case.

We appreciate this valuable feedback from the referee. One of the main achievements of our work is the implication it has for the origin of superconductivity in real materials. We have experimentally shown that magnetically mediated pairing exists in Hubbard-like systems. This emphasises the relevance of magnetic correlations as a potential origin for the pairing underlying high-Tc phases, which has been a topic of debate for the last 35 years.

For our system size and experimental cycle time, the experiment has no direct time advantage over theoretical calculations. However, it is straight forward to expand the measurements to two-dimensional systems, where such an advantage exists. More importantly, the experiment confirms that the effective mixD description is accurate. This is a non-trivial statement as higher order processes in the many-body system could modify the effective Hamiltonian significantly at a finite U/t and Δ/t . We hope our demonstration also motivates material scientists to engineer materials with high binding energy by following the mixD route, and thus potentially reach higher critical temperature.

References

- [1] Bohrdt, A., Homeier, L., Bloch, I., Demler, E. & Grusdt, F. Strong pairing in mixed dimensional bilayer antiferromagnetic Mott insulators. *Nat. Phys.* **18**, 651 (2021).
- [2] Zhang, S.-S., Zhu, W. & Batista, C. D. Pairing from strong repulsion in triangular lattice Hubbard model. *Phys. Rev. B* **97** 140507 (2018).
- [3] Song, Z.-D. & Bernevig, B. A. Matbg as topological heavy fermion: I. exact mapping and correlated insulators *arXiv:2111.05865* (2021).
- [4] Blumberg, G. *et al.* Sliding Density Wave in $\text{Sr}_{14}\text{Cu}_{24}\text{O}_{41}$ Ladder Compounds. *Science* **297**, 584 (2002).
- [5] Golden, M. S. *et al.* The electronic structure of cuprates from high energy spectroscopy. *J. Electron Spectrosc. Relat. Phenom.* **117**, 203–222 (2001).
- [6] Chaudhary, S., Hsieh, D. & Refael, G. Orbital floquet engineering of exchange interactions in magnetic materials. *Phys. Rev. B* **100**220403 (2019).

Reviewer Reports on the First Revision:

Referees' comments:

Referee #1 (Remarks to the Author):

The authors answered all my questions and implemented all corrections I requested. I therefore recommend the publication of the present manuscript.

Referee #2 (Remarks to the Author):

I read the answers of the authors to the issues brought forward by all three reviewers and the new submitted manuscript. I admire the authors for making considerable revisions. These results will be of value to the physics community. My stance is unchanged from before, and in conclusion, I would say that the manuscript warrants publication in Nature.

Referee #3 (Remarks to the Author):

The authors have addressed my questions regarding finite size effects, temperature, and added a very useful concluding paragraph on the implications for real superconductors. I am pleased to give my strong recommendation that this paper be published in Nature.

Randall Hulet